# Size-Dependent Solute Segregation at Symmetric Tilt Grain Boundaries in *α*-Fe: A Quasiparticle Approach Study

**DOI:** 10.3390/ma14154197

**Published:** 2021-07-27

**Authors:** Helena Zapolsky, Antoine Vaugeois, Renaud Patte, Gilles Demange

**Affiliations:** GPM, UMR CNRS 6634, Université de Rouen-Normandy, 76575 Saint Étienne du Rouvray, France; antoine.vaugeois@univ-rouen.fr (A.V.); renaud.patte@univ-rouen.fr (R.P.)

**Keywords:** grain boundary, quasiparticle approach (QA), segregation, phase-field crystal (PFC)

## Abstract

In the present work, atomistic modeling based on the quasiparticle approach (QA) was performed to establish general trends in the segregation of solutes with different atomic size at symmetric 〈100〉 tilt grain boundaries (GBs) in *α*-Fe. Three types of solute atoms X_1_, X_2_ and X_3_ were considered, with atomic radii smaller (X_1_), similar (X_2_) and larger (X_3_) than iron atoms, respectively, corresponding to phosphorus (P), antimony (Sb) and tin (Sn). With this, we were able to evidence that segregation is dominated by atomic size and local hydrostatic stress. For low angle GBs, where the elastic field is produced by dislocation walls, X_1_ atoms segregate preferentially at the limit between compressed and dilated areas. Contrariwise, the positions of X_2_ atoms at GBs reflect the presence of tensile and compressive areal regions, corresponding to extremum values of the *σ_XX_* and *σ_YY_* components of the strain tensor. Regarding high angle GBs Σ5 (310) (*θ* = 36.95°) and Σ29 (730), it was found that all three types of solute atoms form Fe_9_X clusters within B structural units (SUs), albeit being deformed in the case of larger atoms (X_2_ and X_3_). In the specific case of Σ29 (730) where the GB structure can be described by a sequence of |BC.BC| SUs, it was also envisioned that the C SU can absorb up to four X_1_ atoms vs. one X_2_ or X_3_ atom only. Moreover, a depleted zone was observed in the vicinity of high angle GBs for X_2_ or X_3_ atoms. The significance of this research is the development of a QA methodology capable of ascertaining the atomic position of solute atoms for a wide range of GBs, as a mean to highlight the impact of the solute atoms’ size on their locations at and near GBs.

## 1. Introduction

The physical properties of polycrystalline materials (embrittlement, electric transport, and corrosion) are often driven by the segregation of impurities at grain boundaries (GBs) [1,2,3,4,5]. Manifold experimental studies have been carried out to investigate the interaction of impurities with GBs [6,7,8,9,10]. Nowadays, it is well established that sulfur (S) and phosphorous (P) atoms are embrittling elements in steel, while carbon (C) and boron (B) are known as flagship GB cohesion enhancers [11].

In addition to experiments, computer simulation of GB segregation have ushered in atomic level investigations of the segregation process. In this regard, significant progress has been made to decipher the interaction between solute atoms and GBs by means of molecular dynamic (MD) and ab initio calculations [12,13,14,15,16,17]. Ab-initio methods are frequently used to evaluate the most stable segregation sites. For instance, J. Wang et al. [18] looked into segregation at symmetric tilt grain boundaries in *α*-Fe. It was evidenced that the maximum fracture strength of a GB depends on the maximum carbon concentration that can be accommodated by these GBs. Y. Hu et al. [19] studied the segregation effects of six transition metal elements (Cr, Ni, Cu, Zr, Ta, and W) on the Σ3 (111) tilt boundary in bcc iron and concluded that the segregation of Zr, Ni and Cu elements decreases the GB cohesive strength, as opposed to the strengthening effect of Cr, Ta, and W atoms. Ab initio calculations also entailed meaningful progresses regarding the comprehension of the phenomenon of hydrogen embrittlement of metals and alloys [20,21]. Despite paramount breakthroughs in the field of GB interactions with solute atoms enabled by ab initio calculations, this class of approaches remains computationally expensive, hence its restricted use to a few GBs at 0 K. Alternatively, MD modeling spearheaded a host of new results on GBs segregation. For example, N. R. Rhode et al. [22] examined a segregation of carbon atoms on a large number of GBs, including general low and high angle GBs. In particular, they underlined the influence of the local structure of the GB on the segregation energy. The main result of this work was to demonstrate that the atomic sites at and close to GBs show an asymmetric distribution of segregation energies displaying extreme values that extend over 10 Å from the grain boundary. Albeit significantly less computationally demanding than the ab initio calculations, the MD simulation is limited to the description of systems on relatively short time scales, so it struggles to reproduce the diffusion of solute atoms toward GB. Notwithstanding this shortcoming, MD modeling allowed to gather a comprehensive database for the segregation spectra of 250+ binary alloys [23], which was successfully exploited in the machine learning framework to predict the segregation energy of a solute atom in a GB site. However, new methods are now required to improve this database and provide insight on the full segregation phenomenon, starting from the solute diffusion to GB, up to equilibrium segregation at GBs.

In this regard, an orthogonal approach to MD is provided by the phase-field crystal (PFC) model [24,25,26]. This approach was first curtailed to bcc symmetry, and thereupon expanded to arbitrary crystallography [27] under the banner of XPFC [28]. Despite the great success of the XPFC method in describing GB structures [29], this approach is currently not suitable to account for atoms of different radii, which hinders the assessment of the solute atoms radius influence on GB segregation patterns. This shortcoming can be circumvented by an alternative model to XPFC, coined the quasiparticle approach (QA). This method was the outcome of the extension of the atomic density function approach (ADF) [30] to the continuum case [31]. With this, it opened a way to model the evolution of different aperiodic systems (such as GBs or glass) or displacive phase transformations. The QA has already been applied to model the symmetric tilt grain boundaries structure [32] and vacancies annihilation [33,34] at GBs in *α*-Fe, the self-assembly of atoms into complex structures [35], and fcc/bcc phase transformations [36].

In this work, we propose to use the QA to study the segregation of solute atoms at symmetric 〈100〉 tilt GBs in *α*-Fe. The main objective of this paper is to provide insight on how the local structure of GB can affect the segregation of various elements with different atomic radii. Specifically, the present work aims at disentangling the strain and chemical contributions of solute atoms segregation at GBs, upon modulating the atomic radii of solute atoms, while keeping the chemical interactions fixed. For this purpose, three types of solute atoms with different atomic radii and two classes of GBs were considered: low angle GBs (LAGBs), with *θ* = 7.15° and *θ* = 9.53°, and two high angle GBs (HAGBs), Σ5 (310) (*θ* = 36.95°) and Σ29 (730) (*θ* = 46.40°). This choice was motivated by the fact that Σ5 (310) is a low fit index symmetric (special) GB in the coincident site lattice (CSL) theory [37] characterized by a low segregation tendency [4,38]. Contrariwise, Σ29 (730) is a high index (general) GB, which displays a high segregation tendency. Finally, LAGBs (*θ* < 15°) display a dislocation wall structure [39].

This paper is organized as follows. First, an overview of the quasiparticle approach with application to the binary system is presented, with special emphasis on the choice of model parameters. The QA is then applied to model the solute diffusion and segregation at symmetric 〈100〉 tilt GBs in *α*-Fe. Finally, we compare our results with the available ab initio calculations. We demonstrate that the proposed model gives a good description of the solute atoms distribution at the considered GBs in *α*-Fe.

## 2. Quasiparticle Approach (QA)

### 2.1. Basic Equation

In this section, the main equations of the quasiparticle approach (QA) are briefly introduced. This method can be seen as extension of the seminal atomic density function (ADF) approach [30] to the continuum case. In the ADF theory, the atomic configuration is described by a set of occupation probabilities $P_{\alpha }\left(r,t\right)$, defined as the probability for a given lattice site r to be occupied by the atom of type α, at a given time *t*. The temporal evolution of these variables is governed by Onsager-type diffusion equations, wherein it is proportional to the thermodynamic driving force [30]. In the ADF approach, the probability function is specified at each site of the underlying Ising lattice, which coincides with the simulation grid, thereby confining the application range of the model to isostructural phase transformations. This shortcoming was later circumvented in [31] upon choosing the simulation grid spacing several times smaller than the interatomic distance. With this, the so-called continuum atomic density function (CADF) theory allowed to account for the atomic movement in the continuous space, hence making the model applicable to structural phase transitions. In the CADF framework, atoms are no longer points, but rather spheres containing a certain number of the simulation grid nodes (see Figure 1). In this regard, a new interaction Hamiltonian should be defined to set the dynamics of the system. One such Hamiltonian was proposed in an upgrade of the CADF model referred to as the quasiparticle approach (QA) [35].

The salient feature of this second extension of the ADF theory relies on the treatment of lattice points belonging to atomic spheres as non-traditional dynamic variables called fratons. In the QA, the configurational degrees of freedom are occupation numbers c(r). The function c(r) is equal to 1 if the lattice site in position r lies within the atomic sphere, and is 0 otherwise. In this formalism, the occupation numbers of the fratons are dynamic variables of the system rather than the coordinates of atomic spheres in the conventional description of the configurational phase space. Therefore, a *m*-components system can be characterized by the values of *m* stochastic numbers $c_{\alpha }\left(r\right)$ in each lattice site r, where α=1,2,..m labels the fratons related to the corresponding atom of atomic species α. Occupation variables $c_{\alpha }$ are then averaged over the time-dependent Gibbs ensemble into the occupation probability $\rho_{\alpha }\left(r,t\right)=\langle c_{\alpha }\left(r,t\right)\rangle$, where the 〈·〉 symbol denotes the Gibbs ensemble average at temperature *T* and time *t*. With this definition, the function $\rho_{\alpha }\left(r,t\right)$ is the probability that a lattice point in r is located anywhere inside the atomic sphere of any atom of the kind α at time *t*. Therefore, the atomic configuration of the system can be fully described by the density function $\rho_{\alpha }\left(r,t\right)$. Moreover, the temporal evolution of the system is given by the microscopic diffusion equation [31]:(1)$$\frac{\partial \rho_{\alpha }}{\partial t}\left(r,t\right)=\sum_{\beta =1}^{m}\sum_{r^{\prime }\in I}L_{\alpha \beta }\left(r-r^{\prime }\right)\frac{\delta F}{\delta \rho_{\beta }\left(r^{\prime },t\right)}$$
where the summation is carried out over the $N_{0}$ grid sites of the Ising lattice *I*. In Equation (Equation 1), $L_{\alpha \beta }\left(r-r^{\prime }\right)$ is the matrix of kinetic coefficients, where indices α and β label fratons describing the different kinds of atoms (α=1,2,m), and *F* is the non-equilibrium Helmholtz free energy functional. Kinetic Equation (Equation 1) approximates the evolution rate of the density functions $\rho_{\alpha }\left(r,t\right)$ by the first non-vanishing term of its expansion with respect to the thermodynamic driving force in the small driving force limit. This microscopic diffusion equation is significantly nonlinear with respect to the density field $\rho_{\alpha }\left(r,t\right)$, but it is linear with respect to the driving force. To guarantee a conservation of the total number of fratons of the kind α, the kinetic coefficients matrix should satisfy the following condition for all α,β=1,…,m:(2)$$\sum_{r\in I}L_{\alpha \beta }\left(r\right)=0.$$

As for the free energy *F* of the system, it is defined under mean-field approximation by the following:(3)$$\begin{array}{cc} F & =\sum_{\alpha =1}^{m}\sum_{\begin{array}{c} \beta =1\\ \beta \geq \alpha \end{array}}^{m}[\frac{1}{2}\sum_{r,r^{\prime }\in I}W_{\alpha \beta }\left(r-r^{\prime }\right)\rho_{\alpha }\left(r,t\right)\rho_{\beta }\left(r^{\prime },t\right)]\\ & +k_{B}T\sum_{r\in I}[\sum_{\alpha =1}^{m}\rho_{\alpha }\ln\left(\rho_{\alpha }\right)+\left(1-\sum_{\alpha =1}^{m}\rho_{\alpha }\right)\ln(1-\sum_{\alpha =1}^{m}\rho_{\alpha })], \end{array}$$
where $k_{B}$ is the Boltzmann constant and *T* is the temperature. In this expression, the first term on the right of the equality corresponds to the internal energy, while the term on the right stands for the configurational entropy. $W_{\alpha \beta }$ is the pairwise interaction potential between fratons of type α and β separated by a distance $|r-r^{\prime }|$. It is noteworthy that the mean-field approximation [38] posited in Equation (Equation 3) is asymptotically accurate at low and high temperatures, while its precision peaks when the interaction radius largely exceeds the distance between interacting particles [38]. Thus, the smaller the grid spacing, the more accurate the description of the continuous atomic movements. However, refining the simulation lattice feeds through an increment of the computational cost, so a trade-off between a fine simulation and computational efficiency should be reached.

In the QA, the model fraton–fraton pair potential $W_{\alpha \beta }$ embodies the so-called short range (SR) and long range (LR) interactions, respectively written $\theta_{\alpha }$ and $W_{\alpha \beta }^{\mathrm{LR}}$:(4)$$W_{\alpha \beta }\left(r-r^{\prime }\right)=\theta_{\alpha }\left(r-r^{\prime }\right)\delta_{\alpha \beta }+\lambda_{\alpha \beta }W_{\alpha \beta }^{\mathrm{LR}}\left(r-r^{\prime }\right).$$

Here, $\delta_{\alpha \beta }$ is the Kronecker delta function. The parameter $\lambda_{\alpha \alpha }$ (α=β) is the relative amplitude between LR and SR interactions. In this work, we set $\lambda_{\alpha \alpha }\equiv \lambda$ for all components α=1,…,m. No SR cross interactions (α≠β) are considered, so $W_{\alpha \beta }\equiv \lambda_{\alpha \beta }W_{\alpha \beta }^{\mathrm{LR}}$. The SR contribution $\theta_{\alpha }$ allows the spontaneous condensation of fratons into atomic spheres, and prevents the overlap or atoms. To reproduce this specific behavior of fratons at short distances, the step function depicted in Figure 2 was used for SR interactions [35]:(5)$$\theta_{\alpha }\left(r\right)=\{\begin{array}{c} -1\;\mathrm{if}\;r\leq R_{\alpha }\\ \xi \;\mathrm{if}\;R_{\alpha }<r\leq R_{\alpha }+\Delta R_{\alpha }\\ 0\;\mathrm{otherwise}, \end{array}$$

Here, $R_{\alpha }$ sets the width of the attractive part of the SR potential, which determines the radius of atomic spheres for component α. Then, $\Delta R_{\alpha }$ and ξ are respectively the width and height of the SR potential barrier for each component α. The introduction of a repulsive contribution in the SR potential not only prevents the overlap of atomic spheres, but also contributes to adjust the elastic properties of the system. For convenience, the interaction potential $W_{\alpha \beta }$ is implemented in reciprocal space by means of the Fourier transforms ${\hat{\theta }}_{\alpha }\left(k\right)$ and ${\hat{W}}_{\alpha \beta }^{\mathrm{LR}}\left(k\right)$ of the SR and LR interactions, where k is the k-vector defined by $k=\left(k_{x},k_{y},k_{z}\right)=\left(\frac{2\pi }{N_{x}}h,\frac{2\pi }{N_{y}}k,\frac{2\pi }{N_{z}}l\right)$ with (h,k,l) being dimensionless coordinates, and $N_{x}$, $N_{y}$ and $N_{z}$ being the number of grid nodes on each edge of the simulation domain. Equation (Equation 4) thus becomes the following:(6)$${\hat{W}}_{\alpha \beta }\left(k\right)={\hat{\theta }}_{\alpha }\left(k\right)\delta_{\alpha \beta }+\lambda_{\alpha \beta }{\hat{W}}_{\alpha \beta }^{\mathrm{LR}}\left(k\right),$$
where the Fourier transformation ${\hat{\theta }}_{\alpha }\left(k\right)$ of the SR contribution reads as follows:(7)$$\begin{array}{cc} {\hat{\theta }}_{\alpha }\left(k\right) & =\frac{4\pi }{k^{3}}[-\sin\left(kR_{\alpha }\right)+kR_{\alpha }\cos\left(kR_{\alpha }\right)+\xi \{\sin(k\left(R_{\alpha }+\Delta R_{\alpha }\right))\\ \; & -k\left(R_{\alpha }+\Delta R_{\alpha }\right)\cos\left(k\left(R_{\alpha }+\Delta R_{\alpha }\right)\right)\\ \; & -\sin\left(kR_{\alpha }\right)+kR_{\alpha }\cos\left(kR_{\alpha }\right)\}]. \end{array}$$

In this work, the segregation of one solute species X at a GB in bcc iron is addressed. Henceforth, the first component (α=1) will conventionally pertain to Fe, while the second component (α=2) will refer to the solute species X. The corresponding occupation probabilities will then be written $\rho_{1}\left(r,t\right)\equiv \rho_{\mathrm{Fe}}\left(r,t\right)$ and $\rho_{2}\left(r,t\right)\equiv \rho_{X}\left(r,t\right)$ in the remaining part of the paper.

The long range interaction potential ${\hat{W}}_{\alpha \beta }^{\mathrm{LR}}\left(k\right)$ defines the crystal structure, the elastic properties, and the chemical interactions between atoms in the system. In this work, only spherically symmetric potentials were used as a means to allow the formation of crystallographic structures with arbitrary orientation. The potential ansatz introduced in [31] was selected to describe the (bcc) crystallographic structure and the elastic properties of α-Fe. As was shown in [32] for a single-component system, the LR contribution can be conveniently fitted on the structure factor S(k) of a given system close to the melting point. In this work, the S(k) function for the bcc-iron calculated in [40] was fitted by the following function:(8)$${\hat{W}}_{11}^{\mathrm{LR}}\left(k\right)\equiv {\hat{W}}_{\mathrm{Fe}}^{\mathrm{LR}}\left(k\right)=1-\frac{k^{4}}{{\left(k^{2}-k_{1}^{2}\right)}^{2}+k_{2}^{4}}.$$

Here, $k_{1}$ and $k_{2}$ are two model parameters that notably set the characteristic wavelength $k_{0}=\frac{\sqrt{k_{1}^{4}+k_{2}^{4}}}{k_{1}}$ of the bcc crystal lattice, for the latter value minimizes the function in Equation (Equation 8). In the bcc structure, each atom is surrounded by 8 nearest neighbors, and the first neighbor distance between atoms is given by $d=a_{0}\sqrt{3}/2$, where $a_{0}$ is the lattice parameter of α-iron. Recalling that the maximum allowed radius for an atom embedded in a crystal structure is half the nearest neighbor distance *d*, the following condition for the step functions $\theta_{\mathrm{Fe},X}$ defined in Equation (Equation 5) holds:(9)$$R_{\mathrm{Fe},X}+\Delta R_{\mathrm{Fe},X}\leq a_{0}\sqrt{3}/4.$$

The reciprocal lattice of the bcc crystal is the fcc structure with the reciprocal lattice parameter $4\pi /a_{0}$. It ensues that the first structural reflection is located at $k_{0}=2\sqrt{2}\pi /a_{0}$. As was discussed in [35,41], a multi-minima potential is required to model more complex structures. From a general perspective, the number of minima of such LR potential should be equal to the number of non-equivalent structural reflections in the first Brillouin zone of the system.

Under the dilute solution hypothesis, we assume in this work that ${\hat{W}}_{22}^{\mathrm{LR}}\left(k\right)\equiv {\hat{W}}_{X}^{\mathrm{LR}}\left(k\right)=0$. Moreover, a fully repulsive interaction was considered for the long-range potential ${\hat{W}}_{12}^{\mathrm{LR}}\left(k\right)$ between solute and Fe atoms. This contribution was implemented by a simple Gaussian function centered in k=0 as a means to foster the phase separation of the different chemical species:(10)$${\hat{W}}_{12}^{\mathrm{LR}}\left(k\right)\equiv {\hat{W}}_{\mathrm{Fe}\text{-}X}^{\mathrm{LR}}\left(k\right)=-\exp\left(-\frac{k^{2}}{2\sigma^{2}}\right).$$

The fitting parameter σ tailors the range of the Fe-X repulsion, depending on the solute species X. As for the cross interaction potential, we recall that ${\hat{W}}_{12}=\lambda_{12}{\hat{W}}_{12}^{\mathrm{LR}}$ in Equation (Equation 6). Using the same notations as in Equations (Equation 8)–(Equation 10) (index 1 for Fe, index 2 for X), this relation reads ${\hat{W}}_{\mathrm{Fe}\text{-}X}=\lambda_{\mathrm{Fe}\text{-}X}{\hat{W}}_{\mathrm{Fe}\text{-}X}^{\mathrm{LR}}$. With this, the parameter $\lambda_{\mathrm{Fe}\text{-}X}$ weights the relative influence of the iron structural contribution ${\hat{W}}_{\mathrm{Fe}}$ with respect to the chemical repulsion ${\hat{W}}_{\mathrm{Fe}\text{-}X}^{\mathrm{LR}}$ between iron and solute X atoms.

It should be also pointed out that each grid node can be occupied or not by a fraton. Then, in order to describe the atomic configuration in a binary system, the distribution of Fe, X and vacancies (V) should be considered. However, according to the conservation condition $\rho_{\mathrm{Fe}}+\rho_{X}+\rho_{V}=1$, only two fraton density functions should be defined. With indices 1 and 2 referring to Fe and X atoms, kinetic Equation (Equation 1) for the density probability functions $\rho_{\mathrm{Fe},X}$ was solved in Fourier space:(11)$$\begin{array}{cc} \frac{\partial {\hat{\rho }}_{\mathrm{Fe}}}{\partial t}\left(k,t\right) & ={\hat{L}}_{\mathrm{Fe}}\left(k\right)[{\hat{W}}_{\mathrm{Fe}}\left(k\right){\hat{\rho }}_{\mathrm{Fe}}\left(k,t\right)+{\hat{W}}_{\mathrm{Fe}\text{-}X}\left(k\right){\hat{\rho }}_{X}\left(k,t\right)+k_{B}T{\left\{\ln\left(\rho_{\mathrm{Fe}}/\left(1-\rho_{\mathrm{Fe}}-\rho_{X}\right)\right)\right\}}_{k}]\\ & +{\hat{L}}_{\mathrm{Fe}\text{-}X}\left(k\right)[{\hat{W}}_{X}\left(k\right){\hat{\rho }}_{X}\left(k,t\right)+{\hat{W}}_{\mathrm{Fe}\text{-}X}\left(k\right){\hat{\rho }}_{\mathrm{Fe}}\left(k,t\right)+k_{B}T{\left\{\ln(\rho_{X}/\left(1-\rho_{\mathrm{Fe}}-\rho_{X}\right))\right\}}_{k}]\\ \frac{\partial {\hat{\rho }}_{X}}{\partial t}\left(k,t\right) & ={\hat{L}}_{X}\left(k\right)[{\hat{W}}_{X}\left(k\right){\hat{\rho }}_{X}\left(k,t\right)+{\hat{W}}_{\mathrm{Fe}\text{-}X}\left(k\right){\hat{\rho }}_{\mathrm{Fe}}\left(k,t\right)+k_{B}T{\left\{\ln(\rho_{X}/\left(1-\rho_{\mathrm{Fe}}-\rho_{X}\right))\right\}}_{k}]\\ & +{\hat{L}}_{\mathrm{Fe}\text{-}X}\left(k\right)[{\hat{W}}_{\mathrm{Fe}}\left(k\right){\hat{\rho }}_{\mathrm{Fe}}\left(k,t\right)+{\hat{W}}_{\mathrm{Fe}\text{-}X}\left(k\right){\hat{\rho }}_{X}\left(k,t\right)+k_{B}T{\left\{\ln(\rho_{\mathrm{Fe}}/(1-\rho_{\mathrm{Fe}}-\rho_{X}))\right\}}_{k}], \end{array}$$
where ${\hat{\rho }}_{\mathrm{Fe},X}\left(k,t\right)=\sum_{r}\rho_{\mathrm{Fe},X}\left(r,t\right)\exp\left(ik\cdot r\right)$ is the Fourier transform of the density function $\rho_{\mathrm{Fe},X}$, and ${\left\{\cdot \right\}}_{k}$ is the discrete Fourier transform operator. Moreover, ${\hat{L}}_{\mathrm{Fe},X,\mathrm{Fe}\text{-}X}\left(k\right)=-L_{\mathrm{Fe},X,\mathrm{Fe}\text{-}X}^{\mathrm{Ons}}k^{2}$, where the coefficients $L_{\mathrm{Fe},X,\mathrm{Fe}\text{-}X}^{\mathrm{Ons}}$ are Onsager diffusion coefficients. In the present model, the interaction between vacancies and atoms as well the vacancy–vacancy interactions were neglected. Under this simplification, the Onsager diffusion coefficient matrix reads as follows:(12)$$L^{\mathrm{Ons}}=\left(\begin{array}{cc} L_{\mathrm{Fe}}^{\mathrm{Ons}} & L_{\mathrm{Fe}\text{-}X}^{\mathrm{Ons}}\\ L_{\mathrm{Fe}\text{-}X}^{\mathrm{Ons}} & L_{X}^{\mathrm{Ons}} \end{array}\right).$$

Different values of the coefficients of $L^{\mathrm{Ons}}$ matrix were assessed in this work (not shown). It was observed that the equilibrium state obtained by solving Equation (Equation 11) remains quite insensitive to the exact values chosen for these coefficients, provided that the matrix is positive definite. With this, Onsager coefficients were set to $L_{\mathrm{Fe}}^{\mathrm{Ons}}=L_{X}^{\mathrm{Ons}}=1$ and $L_{\mathrm{Fe}\text{-}X}^{\mathrm{Ons}}=0.5$ in all simulations.

Equation (Equation 11) was solved in reduced units. The average density of probabilities ${\bar{\rho }}_{\mathrm{Fe}}$ and ${\bar{\rho }}_{X}$ of matrix (Fe) and solute (X) atoms is defined as $4\pi R_{\mathrm{Fe},X}^{3}N_{\mathrm{Fe},X}/\left(3V\right)$, where $V={\left(\Delta x\right)}^{3}N_{x}N_{y}N_{z}$ is the total volume of the system, and $N_{\mathrm{Fe}}$ ($N_{X}$) is the total number of Fe (X) atoms at ground state. The input parameters related to the energy ξ, λ and $\lambda_{\mathrm{Fe}\text{-}X}$ are expressed in $k_{B}T_{m}$ units, where $T_{m}$ is the melting temperature of the pure iron system with composition ${\bar{\rho }}_{\mathrm{Fe}}$. Lengths are expressed in units of the lattice parameter Δx, so the grid spacing Δx of the underlying Ising lattice is defined as a fraction of the lattice parameter $a_{0}$ of Fe.

### 2.2. Model Parameters

In this work, the effect of the solute atom radius on the segregation at GBs in α-iron ($a_{0}=2.87$ Å) was investigated. With this in mind, solute atoms X_1_, X_2_ and X_3_ with three different ionic radius tantamount to that of phosphorus (P), antimony (Sb) and tin (Sn) were considered. The corresponding values of $R_{{\mathrm{Fe},X}_{1,2,3}}$ and $\Delta R_{{\mathrm{Fe},X}_{1,2,3}}$ complying with condition (Equation 9) are provided in Table 1. The resulting SR interaction profiles defined in Fourier space in Equation (Equation 7) could then be set, as displayed in Figure 3a.

To fix the model parameters introduced in Equation (Equation 10) for LR interactions between iron atoms, the elastic constants of α-Fe were reckoned, using the following procedure. For small deformations where materials exhibit a linear elastic behavior, the free energy can be expanded in Taylor series with respect to the deformation:(13)$$F\left(\left\{\epsilon_{k}\right\}\right)=F_{0}+\frac{V_{0}}{2}\sum_{m,n=1}^{6}C_{mn}\epsilon_{m}\epsilon_{n},$$
where $\epsilon_{k}$ is the component of the rank 2 strain tensor $\overset{=}{\epsilon }$, $F_{0}$ and $V_{0}$ are the free energy and volume of the unconstrained system respectively, and $C_{mn}$ are the elastic constants in Voigt notation for a cubic system. The latter can be evaluated through the second derivative of the free energy:(14)$$C_{mn}=\frac{\partial^{2}F}{\partial \epsilon_{m}\partial \epsilon_{n}}.$$

To characterize the elastic properties of cubic system, only three independent elastic constants, namely $C_{11}$, $C_{12}$ and $C_{44}$ must be defined. This can be achieved by applying the three characteristic deformations with specific transformation of coordinates to the system:Hydrostatic: (x,y,z)→((1−ϵ)x,(1−ϵ)y,(1−ϵ)z)Orthorhombic: (x,y,z)→((1+ϵ)x,(1−ϵ)y,z)Monoclinic: (x,y,z)→(x+ϵy,y,z)
where the coefficient ϵ is the amplitude of the deformation. It is therefore possible to express the free energy of the system associated with these three deformations as a function of the strain, according to Equation (Equation 13):(15)$$\begin{array}{c} F_{\mathrm{hydro}}=F_{0}+\frac{9}{2}V_{0}B\epsilon^{2}\\ F_{\mathrm{ortho}}=F_{0}+2V_{0}C\epsilon^{2}\\ F_{\mathrm{mono}}=F_{0}+\frac{1}{2}V_{0}C_{44}\epsilon^{2}, \end{array}$$
where $B=(C_{11}+2C_{12})/3$ is the bulk modulus, and $C^{\prime }=\left(C_{11}-C_{12}\right)/2$. Another important parameter called the Zener anisotropy ratio $A_{Z}$ can be estimated from the elastic constants as follows:(16)$$A_{Z}=\frac{2C_{44}}{C_{11}-C_{12}}=\frac{C_{44}}{C^{\prime }}.$$

The case $A_{Z}=1$ corresponds to an isotropic material, whereas $A_{Z}\neq 1$ indicates that the crystal is elastically anisotropic. To evaluate the free energy under three different deformations, a simulation box of $128^{3}$ was used with the next set of model parameters: $a_{0}=16\Delta x$, R=6.15Δx and ΔR=0.7Δx, while different values of ξ in the SR interactions (Equation (Equation 7)) and $k_{1}$ and $k_{2}$ in the LR interactions (Equation (Equation 8)) for α-Fe were analyzed. For ξ=5, $k_{1}=0.435k_{0}$ and $k_{2}=0.626k_{0}$, the following set of elastic constants was found: $C_{11}=163$ GPa, $C_{12}=67$ GPa and $C_{44}=95$ GPa, upon preliminarily fitting the numerical value of $C^{\prime }$ on its ab initio counterpart [42]. The corresponding values of *B* and $A_{Z}$ are 99 GPa and 2.0, respectively. A perfect match with [42] is achieved for $C_{44}$ ($C_{44}=96$ GPa), while the obtained value for $A_{Z}$ falls within the range of numerical (ab initio) and experimental values spanned by the literature ($1.5\leq A_{Z}\leq 2.7$ [42,43,44,45,46]). However, the present bulk modulus *B* is too low with respect to the values reported in the literature (168≤B≤189 [42,43,44,45,46]). This discrepancy for *B* is likely to reflect the small number of free parameters stepping in the expression of LR interactions. One should keep in mind that this underestimation of the bulk modulus of bcc iron in the QA makes the presently modeled iron too soft, compared to experiments. In turn, this may slightly influence the tendency of solute atoms to segregate at GBs depending on their ionic radius. We surmised in the present work that a first qualitative connection between the radius of solute atom and their segregation tendency at GBs could be made. This discrepancy should nonetheless be addressed in a future study, using a more sophisticated interaction potential.

The LR interaction potentials for α-Fe hereby fitted is depicted in Figure 3b (blue), along with the cross interaction potential ${\hat{W}}_{\mathrm{Fe}\text{-}X}^{\mathrm{LR}}\left(k\right)$ (red). In the present work, the same values for $\lambda_{\mathrm{Fe}\text{-}X}$ and σ were used for X_1_, X_2_ and X_3_ in Equation (Equation 10). In this manner, the different solute species only depart from one another through their atomic radius, hence inducing different atomic misfits in the bcc iron lattice. In doing so, we also tacitly ascribe a specific segregation tendency at GBs to atomic the size effect, for purposes of disentangling the chemical and elastic mutual influence on the segregation at GBs. All parameters are compiled in Table 1.

In this work, the specific case of the segregation of one chemical species X at 〈100〉 symmetric tilt grain boundaries with the (010) interface plane in α-Fe was studied using the following procedure. First, the GB structure with specific misorientation angle θ was obtained by crystallizing a liquid layer placed in between two bcc crystal grains rotated by θ/2 around the 〈100〉 axis. In order to find the minimum energy of the system during the crystallization stage, Equation (Equation 1) was integrated until equilibrium was reached. Then solute atoms (X) were introduced in substitutional position with a density of presence of 10%. To satisfy the periodic boundary conditions, two GB were introduced in the simulation box.

The space scale was chosen as the grid spacing Δx=0.018 nm, as imposed by the number of grid lattices (16) spanning one bcc iron lattice parameter $a_{0}=16\Delta x=2.87$ Å. Simulations were performed in three dimensions on a 600×600×64 grid lattice ($N_{x}=N_{y}=600$, $N_{z}=64$) equipped with periodic boundary conditions. For the chosen length scale (Δx≃0.018 nm), the latter corresponds to a volume of (11 nm)^3^. Kinetic Equation (Equation 11) was solved by the spectral-Eyre scheme [41] with the reduced time step Δt=0.005, on the supercomputer CRIANN of Normandy.

## 3. Results

### 3.1. Segregation Pattern at GB

#### 3.1.1. Low Angle GB

In the present work, we have assessed how the size of solute atoms influences the segregation phenomenon, starting with LAGBs. For that purpose, the segregation of X_1_ and X_2_ atoms was investigated at two LAGBs. As a reminder, the radius of X_1_ and X_2_ solute atoms were chosen close to the ionic radius of phosphorus (P) and antimony (Sb), respectively (radius $R_{X_{1}}<R_{X_{2}}$). Tilt symmetrical LAGBs with misorientation angle θ<15° can be described by a wall of edge dislocations [39]. These dislocations alter the elastic field around GB and hereby significantly influence solute atoms diffusion and segregation. In Figure 4, the GB segregation of X_1_ (Figure 4a) and X_2_ (Figure 4b) atoms for the low misorientation angles *θ* = 7.15° and *θ* = 9.53° respectively, are displayed after projection on the (100) plane.

Therein, Fe atoms in two successive (100) planes (*n* and n+1) are colored in white and black respectively. X_1_ solute atoms are observed in (100) Fe planes, as well as (100) Fe interplanes. X_1_ atoms lying in *n*, $n+\frac{1}{2}$ (n/n+1 interplane) and n+1
(100) Fe planes are indicated by red, orange and yellow spheres, respectively. As for X_2_ solute atoms, they only occupy (100) Fe planes. Accordingly, X_2_ atoms lying in *n* and n+1
(100) Fe planes are indicated by red and yellow spheres. In addition, 〈0±10〉 edge dislocations are spotted by red ⊢ marks. On the one hand, the vast majority of X_1_ atoms segregate in interstitial position between two (100) planes of Fe atoms as demonstrated in Figure 4a (orange spheres in $n+\frac{1}{2}$ plane). This observation is consistent with previous ab initio calculations on the segregation of P atoms at symmetric tilt grain boundaries in α-iron [17,47]. Only X_1_ atoms nesting at the dislocation core are situated in (100) Fe planes (red and yellow spheres). On the other hand, X_2_ atoms only segregate in (100) Fe planes (red and yellow spheres in Figure 4b).

**Figure 4 materials-14-04197-f004:**
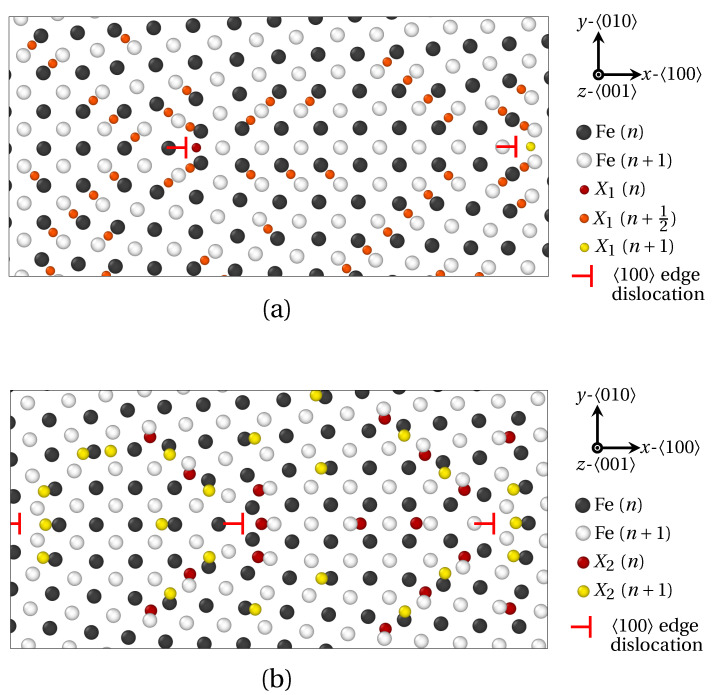
Solute atoms segregation at 〈100〉 symmetric tilt LAGBs consisting of a wall of 〈100〉 edge dislocations (red ⊢ marks), as provided by QA simulations (t=400). (**a**) X_1_ (small) atoms segregation for a misorientation angle θ=7.15°. (**b**) X_2_ (larger) atoms for θ=9.53°. Visualization via OVITO [48] after projection on the (100) plane. Black and white spheres correspond to Fe atoms in two successive (100) planes (*n* and n+1), while red, orange and yellow spheres correspond to solute (X) atoms in three successive (100) planes (*n*, $n+\frac{1}{2}$ and n+1). The size of atoms follows the ordering of solute atoms radius ($R\left(X_{1}\right)<R\left(X_{2}\right)<R\left(X_{3}\right)$).

In both cases, solute atoms segregate around the GB dislocations in interstitial positions, which leads to the formation of so-called Cottrell atmospheres [49] (see Figure 5a for X_1_ atoms and (c) for X_2_ atoms). The latter is widely used to describe carbon atoms segregation at GB in iron [50], as it roots dislocation pinning and dynamic strain aging in steels [51]. This peculiar distribution of solute atoms reflects the stress generated by edge dislocations at GB. This is compressive above the 〈010〉 slip plane (blue shades for atoms on the left of dislocations), and tensile below this plane (red shades on the right of dislocations). With this, solute atoms (orange in Figure 5a,c, green dots in Figure 5b,d) are mainly distributed close to the dislocation core, or at its border, between the dilated and compressed region. Therefore, the periodic pattern for solute atoms segregation stems from the periodic positioning of edge dislocations forming the wall of dislocations at LAGBs.

Complementary information can be deduced from the strain field in the 〈100〉 and 〈010〉 directions at the GB as provided by OVITO (finite strain theory). The strain field is respectively shown in Figure 6a,c for X_1_ atoms, and Figure 6b,d for X_2_ atoms. Albeit X_1_ and X_2_ atoms (green dots) displaying a rather similar segregation pattern in the close vicinity of the dislocation core (⊢ mark) where the dilatation is strong (see Figure 5b,d), they respond differently to the different components of the strain field tensor. First, X_2_ atoms segregate where $\sigma_{XX}<0$ is minimal (blue atoms emphasized by black dashed line in Figure 6b, and $\sigma_{YY}>0$ is maximal (red atoms emphasized by black dashed line in Figure 6d). Contrariwise, X_1_ segregates preferentially at the limit between compressed and dilated areas in the *x* and *y* directions (black dashed line in Figure 6a,c). These departing behaviors between X_1_ and X_2_ atoms is latched to the nature of the interstitial position they are prone to occupy. X_1_ atoms are small enough to be interspersed in 〈111〉 bcc directions. In this regard, this distribution of small solute atoms (X_1_) is reminiscent of crowdions formed by P atoms in irradiated α-Fe notably [52]. As for X_2_ atoms, their radius is too big to be located in the same directions as X_1_ atoms. They rather occupy octahedral interstitial sites. The pivotal factor for their preferential location is thus the tension of the bcc lattice in one direction, so the volume of octahedral sites in the corresponding direction is increased.

#### 3.1.2. High Angle GB

The effect of atomic radius on the position of solute atoms at high angle 〈100〉 symmetric tilt GB is now studied. Two representative misorientation angles were selected: one special GB, Σ5 (310) (*θ* = 36.95°) in Figure 7 and Figure 8, and one general GB, Σ29 (730) (*θ* = 46.40°) in Figure 9 and Figure 10. The structure of HAGBs is usually described using the structural unit (SU) representation [53,54]. In these figures, SUs are highlighted by solid black (plane *n*) and gray (plane n+1) lines. The virgin GB structure of Σ5 (310) is characterized by the structural units |B.B| [55], while the structure of Σ29 (730) consists of a sequence of |BC.BC| SUs.

First, the segregation of X_1_, X_2_ and X_3_ atoms (atomic radius $R_{X_{1}}<R_{X_{2}}<R_{X_{3}}$, comparable to P, Sb and Sn ionic radii, respectively) at Σ5 (310) is addressed, using the same color coding as that for LAGBs. At the GB, each B SU hosts one solute atom only, which lies on the same (100)
α-Fe plane as the first nearest Fe neighbors, disregarding its atomic radius. Now, B units corresponds to the projection on the (100) plane of a three dimensional capped trigonal prisms structure. With this, Fe_9_X clusters are formed upon hosting solute X atoms, as shown in Figure 7b,e,h. This typical structure for Σ5 (310) GB was notably witnessed for phosphorus (same atomic radius as X_1_) and boron segregation in α-iron in previous MD studies [47]. However, while the structure of Fe_9_(X_1_) clusters closely resembles that of Fe_9_P and Fe_9_B (small atomic radius), a significant deviation can be observed for the Fe_9_(X_2_) and Fe_9_(X_3_) cluster structure as displayed in Figure 7c,f,i. Compared to X_1_ atoms, the equilibrium position of X_2_ and X_3_ atoms is shifted to a more off-centered position within their hosting capped trigonal prism. The deformation of the Fe_9_X cluster is maximal for X_3_ atoms (largest atomic radius) in Figure 7i. This migration of X_2_ and X_3_ solute atom within their respective Fe_9_X clusters is accompanied by a deformation of the capped trigonal prism. This is highlighted by the map of displacement vectors of Fe atoms in Figure 7c,f,i, where red arrows stand for Fe atoms displacement (amplified by a factor of 10 in the figure), between the equilibrium structure of virgin GB, and the GB structure after segregation. For small solute atoms (X_1_), the capped trigonal prism is roughly unaffected by segregation, once again in agreement with [47] for atoms with an atomic radius close to P. For bigger atoms, however (X_2_ and X_3_), noticeable displacements of Fe atom can be observed in the (001) plane. The general deformation trend is the dilatation of capped trigonal prism for X_2_ atoms (intermediate atomic radius), versus shear displacements of Fe atoms for X_3_ atoms (largest atomic radius). This observation may suggest that above a certain threshold for solute atoms radius, the SUs accommodate differently the introduction of solute atoms of different sizes.

Interesting features can also be observed in the vicinity of GBs. For small solute atoms (X_1_), a significant segregation can be observed up to five (010) atomic planes on both sides of the GB (see Figure 7c). This observation is consistent with previous molecular static simulations, where a substantially lower segregation energy for C atoms was obtained for distances up to 5 Å away from Σ5
(310), and 10 Å away from Σ29
(730) [22]. A similar pattern can be envisioned for X_2_ atoms located far from the GB (zone 2 in Figure 7d and Figure 8b). However, a narrow depleted zone is present at the immediate border of the GB (zone 1 in Figure 7d and Figure 8b). The same observation applies for the largest solute atoms (X_3_) in Figure 7g and Figure 8c, where the thickness of the depleted region (zone 1) is even increased.

A first interpretation, relying on the volume per atom, can be proposed, based on Figure 8. Generally speaking, a marked dilatation of the host Fe structure at the GB (atoms in red) goes along with an extended compression area bordering the GB (atoms in blue belonging to zones 1 and 2). The dilatation zone reflects the presence of the capped trigonal prism (B units), which are prone to host solute atoms disregarding their size. Now, this dilatation at the GB is balanced by the compression of the bcc Fe lattice in the abutting region. However, the connection between the position of solute atoms and the elastic field in the compressed region depends on the atomic radius of solute atoms in the following manner: small (X_1_) atoms are present in the compressive region, up to a 5–10 Å distance from the GB (green dots in Figure 8a). Intermediate size atoms (X_2_) are absent in the significantly compressed area (zone 1), but present in the moderately compressed area (zone 2). Finally, large atoms (X_3_) are absent almost everywhere in the compressed regions, but some can be spotted in further regions (zone 2 in Figure 8c). Based on this observation, we suggest that small (X_1_) atoms might be less sensible to the compression of the matrix due to their size, hence their recurrent segregation in this area, in spite of the compression. On the contrary, X_2_ and X_3_ atoms remain in octahedral interstitial sites, and may be too big to segregate in regions where the host structure is sufficiently compressed.

In more details, the elastic field at SUs is not rigorously identical in the three cases. Indeed, the dilatation at B SUs is more intense for X_2_ atoms (∼10% in Figure 8b) than for X_1_ and X_3_ atoms (∼7% in Figure 8a,c). This observation is consistent with the nature and the amplitude of the deformation of the capped trigonal prism depicted in Figure 7c,f,i, depending on the atomic radius of the solute atoms: a negligible deformation for X_1_ atoms and a shear deformation for X_3_ atoms, versus a planar dilatation for X_2_ atoms. One consequence of this might be the over compression of the bcc structure in region 1 for X_2_ segregation in Figure 8b, as entailed by the over dilatation at the GB, and a reduced compression in region 2. We surmise that the over compression in region 1 alternatively precludes the segregation of X_2_ atoms in this area, and yet fosters their segregation in region 2 (see Figure 7b and Figure 8b).

One last remark touches upon the periodic distribution of solute atoms in the 〈100〉 direction near the Σ5
(310) GB. This periodicity stems from the structural periodicity of the sequence of B units at the GB. If this conclusion is obvious for solute atoms belonging to the Fe_9_X clusters, we believe that it remains valid in the vicinity of the GB as well. In this case, the structural periodicity at the GB feeds through to the periodicity of the strain deformation tensor close to the GB.

In order to examine the connection between the structural periodicity of the GB and the positions of solute atoms, the segregation of X_1_ and X_2_ atoms at Σ29
(730) (θ=46.40°) was also prospected in Figure 9 and Figure 10. In this case, the majority of the conclusions formulated for Σ5
(310) remain valid for this GB (segregation at and near GBs, preferential interstitial positions for solute atoms, influence of volumetric strain and atomic radius, etc.). Now, notwithstanding an identical position of solute atoms within B units for both Σ5
(310) and Σ29
(730) GB, a different segregation tendency for X_1_ and X_2_ atoms within C units can be observed for Σ29
(730). In this case, four X_1_ atoms are located within C units, versus only one X_2_ atom. Here again, this phenomenon is linked to the smaller atomic radius of X_1_ atoms, which allows the interspersed segregation in less dilated, or even compressed areas of the host structure.

One step further, the Σ29
(730) GB presents a more complicated segregation pattern than Σ5
(310). This likely stems from the presence of two SUs, which generates a more complex stress field in the vicinity of the GB, compared to a GB consisting of B units only. Indeed, it clearly appears in Figure 9b and Figure 10b,c that the periodic segregation of solute atoms reflects the periodic modulation of the strain tensor. These variations of the strain field consists of the alternation of compressed (tensed) areas in the *x* (*y*) direction, with compressed (tensed) areas in the *y* (*x*) direction. The spatial periodicity of this variation of the field is precisely the length of one BC SU in the *x* direction. In detail, X_1_ atoms are preferentially distributed in regions where $\sigma_{XX}$ (Figure 9b) and $\sigma_{YY}$ (not shown) switch signs, while X_2_ atoms circumscribe areas where $\sigma_{XX}\gg 0$ (in red in Figure 10b) and $\sigma_{YY}\ll 0$ (in blue in Figure 10b). This is consistent with the observations of LAGBs.

## 4. Conclusions

In this study, we applied a new atomistic model based on the quasiparticle approach to explore the relationship between the local grain boundary structure and the size of the segregation solute atom in the α-iron crystal. Three types of solute atoms were considered, with three different atomic radii: X_1_ atoms with a radius much smaller than Fe atom (R=0.68R(Fe)) corresponding to phosphorus (P), X_2_ atoms with an atomic radius close to Fe (R=0.93R(Fe)) corresponding to antimony (Sb), and X_3_ with a larger atomic size than Fe (R=1.08R(Fe)) corresponding to tin (Sn). Two cases were investigated: segregation at LAGBs (θ<15°) on the one hand, and at HAGBs Σ5
(310) (θ=36.95°) and Σ29
(730) on the other hand. It was evidenced that all three sorts of atoms segregate at LAGBs in interstitial positions and generate Cottrell atmospheres around GB dislocations. X_1_ and X_2_ atoms respond differently to the different components of the strain field tensor. It was indeed shown that X_1_ (small) solute atoms segregate preferentially at the limit between compressed and dilated areas in the *x* and *y* directions, whereas X_2_ atoms are rather located where $\sigma_{XX}$ is minimal and $\sigma_{YY}$ is maximal (or the opposite). In the case of Σ5
(310) (θ=36.95°) HAGBs, the three types of solute atoms form Fe_9_X clusters in B units, with a capped trigonal prism structure. Upon increasing the size of solute atoms, a dilatation of this capped prism was observed. In detail, X_2_ (larger) solute atoms induce an homogeneous dilatation of the hosting prism, while X_3_ (largest) atoms entail a shear displacements of Fe atoms. One last noteworthy point regarding HAGBs touches upon the presence of a depleted zone at the immediate border of the GB. The width of this area rises with the atomic radius of solute species. This peculiar distribution of solute atoms was shown to nicely reflect the periodic amplitude variations of the elastic field around the GB. In the case of Σ29
(730), which hosts a series of |BC.BC| SUs, a similar segregation trend within B SUs as in Σ5
(310) was observed. However, the position of X_1_ and X_2_ atoms within C SUs is different, insofar as four X_1_ atoms are encountered therein, as opposed to a single X_2_ (or X_3_) atom. In details, X_1_ atoms are preferentially distributed in regions where $\sigma_{XX}$ and $\sigma_{YY}$ switch signs, while X_2_ atoms are located in the areas where $\sigma_{XX}\ll 0$ and $\sigma_{YY}\gg 0$ (or the opposite).

## Figures and Tables

**Figure 1 materials-14-04197-f001:**
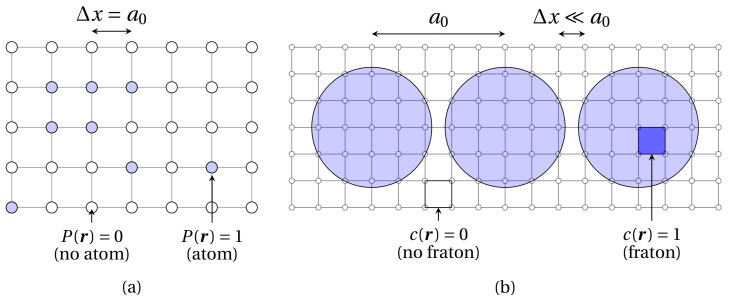
Schematic representation of the simulation grid (gray solid lines connecting grid nodes) for the ADF model (**a**) and the CADF theory and QA (**b**).

**Figure 2 materials-14-04197-f002:**
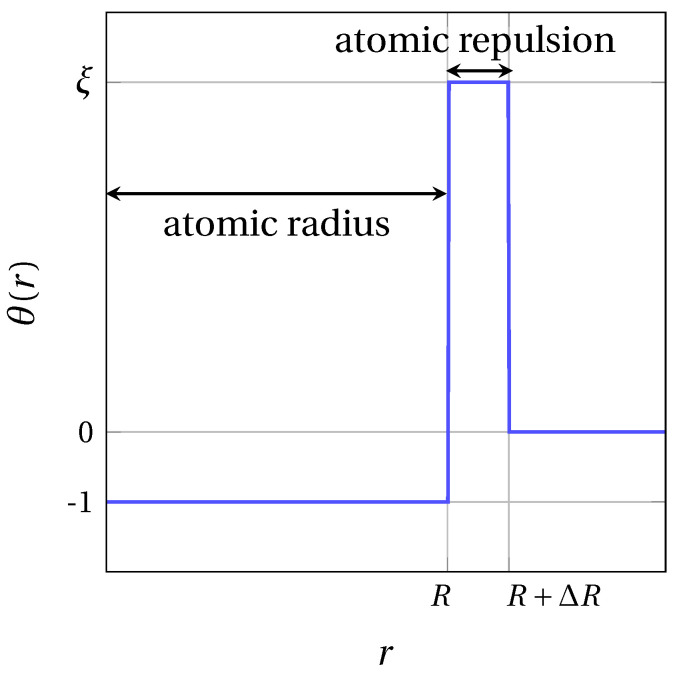
SR interaction $\theta_{\alpha }\left(r\right)$ step function profile.

**Figure 3 materials-14-04197-f003:**
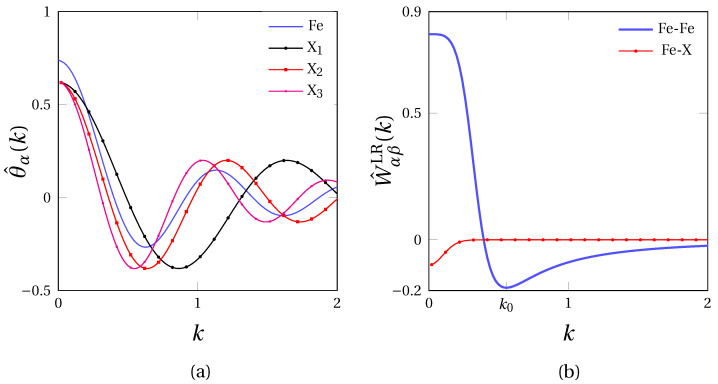
Pairwise interaction potentials used in this work in Fourier space: (**a**) short range interaction potential ${\hat{\theta }}_{\alpha }\left(k\right)$ for Fe, X_1_, X_2_ and X_3_ atoms, (**b**) long range interaction potential ${\hat{W}}_{\mathrm{Fe}}^{\mathrm{LR}}$ (blue) and ${\hat{W}}_{\mathrm{Fe}\text{-}X}^{\mathrm{LR}}$ (red) for X=X_1_, X_2_ and X_3_ using parameters of Table 1.

**Figure 5 materials-14-04197-f005:**
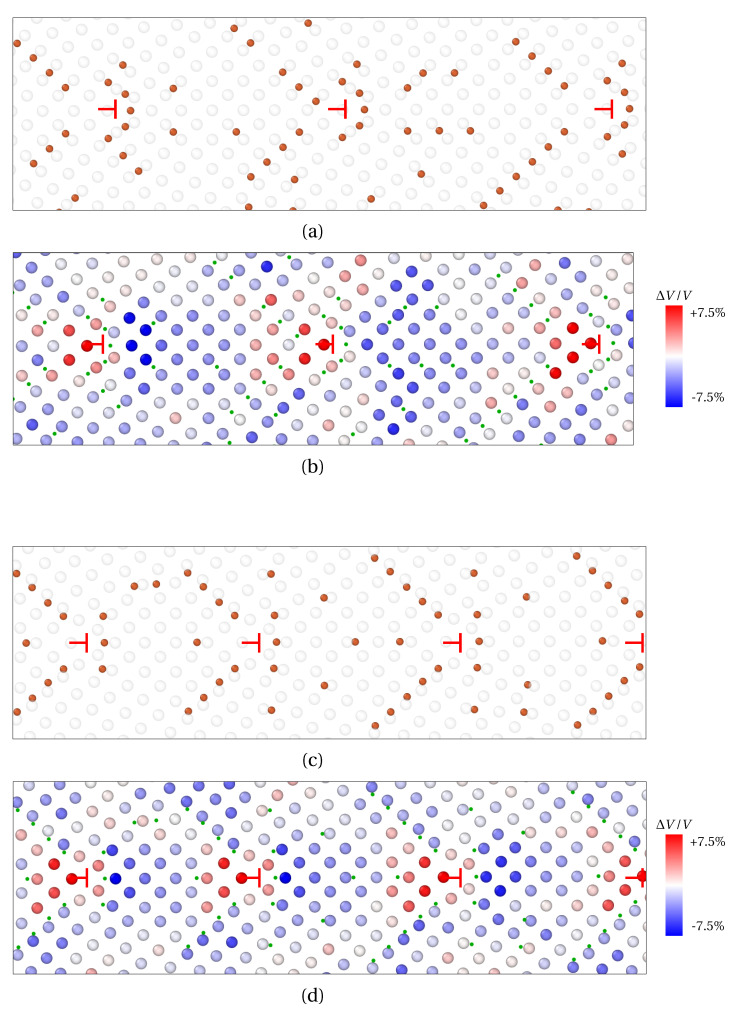
Formation of Cottrell atmospheres at 〈100〉 edge dislocations (red ⊢ marks) after segregation of solute atoms at 〈100〉 symmetric tilt LAGBs. (**a**,**b**) X_1_ (small) atoms segregation for a misorientation angle θ=7.15°. (**c**,**d**) X_2_ (larger) atoms for θ=9.53°. (**a**,**c**) solute atoms distribution (orange) and Fe atoms (transparent). (**b**,**d**) Volume per atom variation ΔV/V (Voronoi analysis). Red—dilatation, blue—compression. X atoms are spotted by green dots in (**b**,**d**).

**Figure 6 materials-14-04197-f006:**
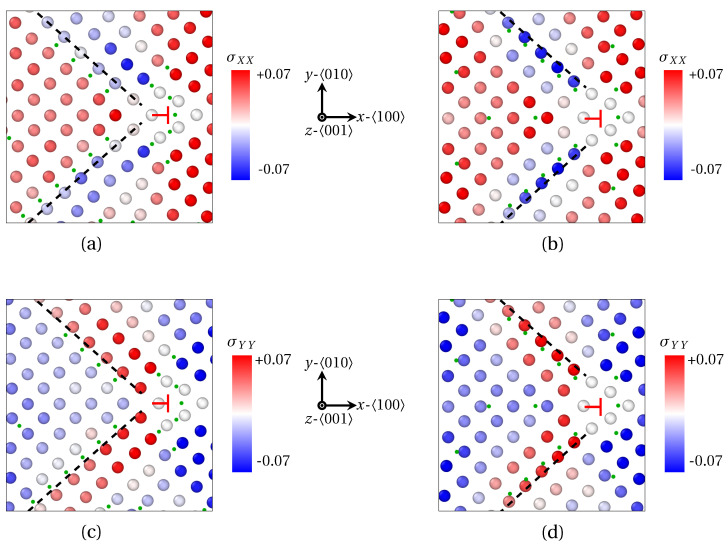
Strain field in the vicinity of one 〈100〉 edge dislocations after segregation of X_1_ (**a**,**c**) and X_2_ (**b**,**d**) atoms at a low angle GB (θ=7.15° for X_1_ and θ=9.53° for X_2_, as provided by QA simulations (t=400). (**a**,**b**) Strain field $\sigma_{XX}$ in the 〈100〉 direction (red: tension in the *x* direction, blue: compression in the *x* direction). (**c**,**d**) Strain field $\sigma_{YY}$ in the 〈010〉 direction for X_2_ (red—tension in the *y* direction, blue—compression in the *y* direction). X atoms are spotted by green dots for the sake of clarity. Black dashed lines: guide for the eye for the preferential area of segregation.

**Figure 7 materials-14-04197-f007:**
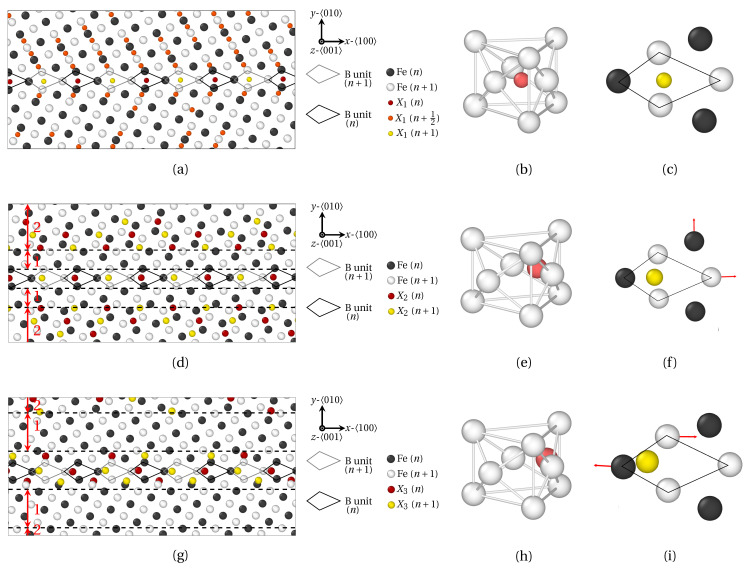
Segregation of solute X_1_ (**a**–**c**) X_2_ (**d**–**f**) and X_3_ (**g**–**i**) atoms at Σ5
(310) (θ=36.87°) as provided by QA simulations (t=400). B structural units are materialized by black (*n* plane) and gray (n+1 plane) lines. (**a**,**d**,**g**) Fe and X atoms positions. (**b**,**e**,**h**) Fe_9_X clusters. (**c**,**f**,**i**) Variation of X solute atom position within B unit, or equivalently Fe_9_X cluster. The depleted (over-compressed) and segregated (less compressed) areas are delineated by dashed black lines, and referred to as zones 1 and 2 in (**a**,**d**,**g**).

**Figure 8 materials-14-04197-f008:**
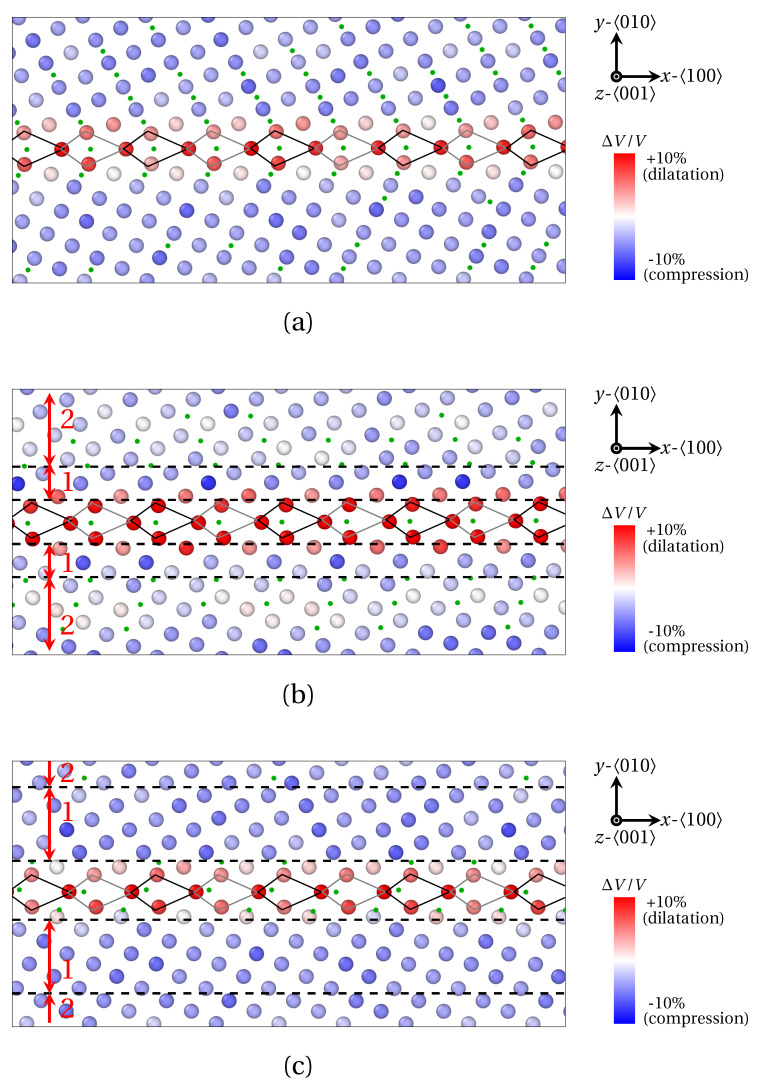
Influence of the volume per atom (Voronoi analysis) on the segregation of X_1_ (**a**) X_2_ (**b**) and X_3_ (**c**) solute atoms at Σ5
(310) (θ=36.87°). X_1_ atoms are spotted by green dots. The depleted (over-compressed) and segregated (less compressed) areas are delineated by dashed black lines, and referred to as zones 1 and 2 in (**b**,**c**).

**Figure 9 materials-14-04197-f009:**
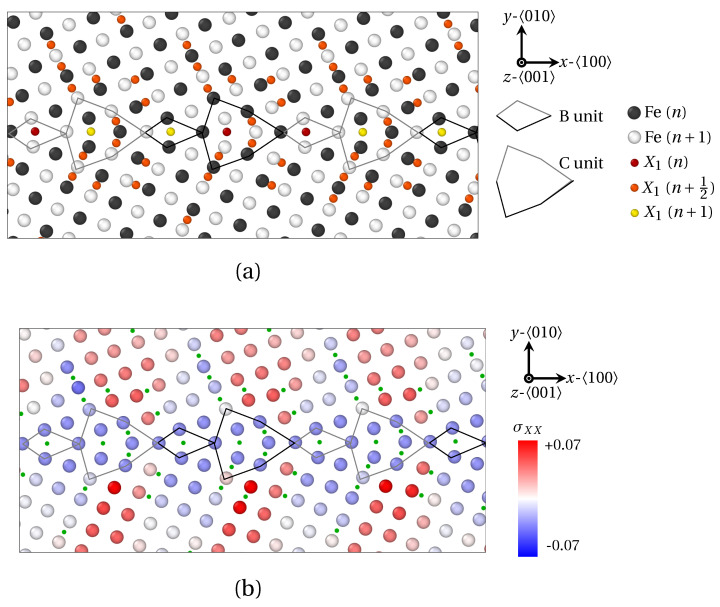
Segregation of X_1_ atoms at Σ29
(730) (θ=46.40°) as provided by QA simulations (t=400). (**a**) Atoms positions. (**b**) Strain field of Fe atoms in the 〈100〉 direction. X_1_ atoms are spotted by green dots. B and C structural units are materialized by black (*n* plane) and gray (n+1 plane) lines.

**Figure 10 materials-14-04197-f010:**
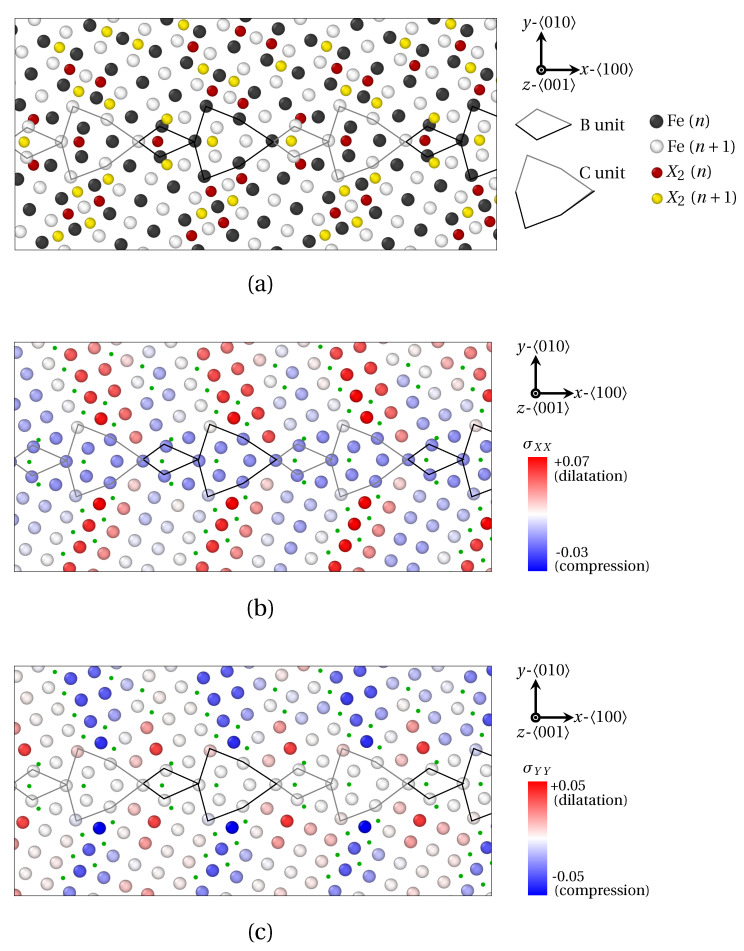
Segregation of X_2_ atoms at Σ29
(730) (θ=46.40°) as provided by QA simulations (t=400). (**a**) Atoms positions. (**b**) Strain field of Fe atoms in the 〈100〉 direction. (**c**) Strain field of Fe atoms in the 〈010〉 direction. X_2_ atoms are spotted by green dots. B and C structural units are materialized by black (*n* plane) and gray (n+1 plane) lines.

**Table 1 materials-14-04197-t001:** Parameters used in this work for QA simulations.

	$a_{0}\left(\Delta x\right)$	R(Δx)	ΔR	ξ	λ	$k_{0}$	$k_{1}$	$k_{2}$	λ	$\lambda_{\mathrm{Fe}\text{-}X}$	σ	$k_{B}T$	$\bar{\rho }$	$L_{\mathrm{Fe}}^{\mathrm{Ons}}$	$L_{\mathrm{Fe}\text{-}X}^{\mathrm{Ons}}$
Fe	16	6.15	1.17R	5.0	0.5	0.555	0.242	0.348	0.5	0.1	0.1	0.0275	0.104	1.0	0.5
X_1_	-	4.2	1.17R	3.0	0.5	-	-	-	0.5	0.1	0.1	0.0275	0.02	1.0	0.5
X_2_	-	5.7	1.17R	3.0	0.5	-	-	-	0.5	0.1	0.1	0.0275	0.02	1.0	0.5
X_3_	-	6.64	1.17R	3.0	0.5	-	-	-	0.5	0.1	0.1	0.0275	0.02	1.0	0.5

## Data Availability

All data sets generated in the current study are available from the corresponding authors upon reasonable request.

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
