# Peer review of "Size-Dependent Solute Segregation at Symmetric Tilt Grain Boundaries in α-Fe: A Quasiparticle Approach Study"

_materials, 2021, doi:10.3390/ma14154197_

Round 1
Reviewer 1 Report
The manuscript by Zapolsky et al. provides an elegant presentation of the size dependence solute segregation to grain boundaries.
The theory, computation, and presentation of the technical content are fully acceptable.
Additional attention to the typographical errors and mistakes in the English grammar and usage must be addressed before this manuscript can be considered for publication. In addition, words such as "prospected" or "bosom" should be replaced with suitable technical language.
Author Response
We would like to sincerely thank the referee for reviewing the present manuscript. Following the referee’s recommendation, a thorough proofread of the manuscript was performed. In this regards, typographical errors and mistakes in the English grammar and usage were systematically corrected. In particular, the word “bosom” was improperly used and was completely removed from the manuscript. The same was done with the verb “prospect”. With this, we believe that the manuscript reads much better now.
Reviewer 2 Report
In this manuscript authors propose to use the quasiparticle approach to study the segregation of solute atoms at symmetric h100i tilt GBs in a-Fe. The information shown in the manuscript are interesting and the paper is well prepared.
In my opinion there is nothing to improve and the paper could be accept in present form.
Regards
Author Response
We would like to sincerely thank the referee for reviewing the present manuscript. We are also very pleased that the reviewer was fully satisfied with our work.
Reviewer 3 Report
The authors present an interesting theoretical study of segregation of impurities/solutes towards grain boundaries in iron. The research is clearly motivated and the authors use their own state-of-the-art method, the Quasiparticle Approach. The simulations are very impressive and the results interesting and very well discussed but a few typos need to be corrected.
For example, I think that the “Önsager-type diffusion” should be written as “Onsager-type diffusion”. Similarly, “Voronoï analysis” should be “Voronoi analysis”. There are some words that probably contain typos, such as “seminal”, “inucing”, “method in describing”, “of GBs. different”, “significInterseting”, “vicinity of GB. ant segregation”, “an identical positions”, “This consistent with the” or “Hereby,it was”.
What I do not understand at all is the term “bosom of the GB”. Can the authors explain what is meant here? It is used about three times throughout the manuscript and while I do study grain boundaries I have never heard about the “bosom of the GB”.
Now, some statements are quite controversial. Having the bulk modulus of iron equal to 99 GPa, well, it is indeed a gross underestimation to about one half of one of the most fundamental elastic constants. Here I do not agree with the statement that “Actually, this issue is also a usual problem in ab-initio calculations, despite the accurate calculations the energy of system.” The problem is that the resulting segregation tendencies of solutes with different size depend on the elastic properties of the iron matrix and I wonder what is the impact of so severly underestimated bulk modulus on the results. Can the authors estimate it, please?
Next, is is indeed so that the published range of Zener ratio is 1.5 < AZn < 2.7 ? Well, that is indeed a very wide range. By the way, why to call the Zener ratio A_Zn (when Zn is a chemical element!!)? It can be just A_Z without any possible confusion ...
In general, the first letter of chemical elements, like iron or carbon, are not capitalized (not "Iron" and "Carbon") despite of the fact that their symbols are capitalized (i.e., Fe, C). Please correct throuhout the whole manuscript.
Next, if the authors set the volume and other parameters of segregating elements “X1, X2 or X3” so that they are matching properties of specific elements such as tin, antimony or sulphur, I strongly suggest that this match to specific elements is mentioned in the Conclusions where the segregation behavior of these elements is discussed. What is this study good for when I only learn that "element X1" is diffusing faster than an "element X2"?
Lastly, as a minor detail, I suggest to call “the supercalculator CRIANN of Normandy” rather a “supercomputer” than a “supercalculator”.
To summarize, I think that this brilliant manuscript needs a minor revision before it can be accepted and published.
Author Response
We would like to sincerely thank the referee for reviewing the present manuscript. All recommendations were addressed in the new version of the manuscript, and further concerns of the referee were also answered in this text. From a general perspective, a thorough proofread of the manuscript was performed. We believe that the manuscript reads much better now. In details:
-A few typos need to be corrected. For example, I think that the “Önsager-type diffusion” should be written as “Onsager-type diffusion”. Similarly, “Voronoï analysis” should be “Voronoi analysis”. There are some words that probably contain typos, such as “seminal”, “inucing”, “method in describing”, “of GBs. different”, “significInterseting”, “vicinity of GB. ant segregation”, “an identical positions”, “This consistent with the” or “Hereby,it was”. What I do not understand at all is the term “bosom of the GB”. Can the authors explain what is meant here? It is used about three times throughout the manuscript and while I do study grain boundaries I have never heard about the “bosom of the GB”.
All these mistakes, typos, and grammatical errors were addressed in the new version of the manuscript, and a thorough proofreading was done. In particular, the word “bosom” was indeed improperly used in the original manuscript and was completely removed from the new version of the paper.
-Now, some statements are quite controversial. Having the bulk modulus of iron equal to 99 GPa, well, it is indeed a gross underestimation to about one half of one of the most fundamental elastic constants. Here I do not agree with the statement that “Actually, this issue is also a usual problem in ab-initio calculations, despite the accurate calculations the energy of system.” The problem is that the resulting segregation tendencies of solutes with different size depend on the elastic properties of the iron matrix and I wonder what is the impact of so severely underestimated bulk modulus on the results. Can the authors estimate it, please ?
Here, the referee is right, the current functional form of the long range (LR) potential (equation 8) leads to a significant underestimation of the bulk modulus for iron, which in turn might influence the segregation tendency of solute atoms. In the new version of the manuscript, this shortcoming was emphasized, in order to warn the reader on this problem. To summarize, it was mentioned in the new version of the manuscript that if the bulk modulus is underestimated as it is the case in the present work, the displacement of solute atoms can produce larger displacements of the host atoms. In addition to what was added to the manuscript, we also would like to argue that the present QA simulations provided a distribution of small solute atoms at the GB (X_1 atoms) which was in good agreement with previous MD studies (see [48] in the article) on the segregation of solute atoms with similar atomic radius as X_1 (phosphorus). Actually, the small value of the bulk modulus for iron in the present work could have a more detrimental impact on the segregation pattern of bigger atoms such as X_3. In this case, the segregation of solute atoms might cause a larger stress on the iron structure than what was predicted in the present work. However, it should be mentioned that previous DFT calculations (see [1] just below) provided a similar structure for the Fe_9X cluster for Sn atoms, which have a similar atomic radius to X_3.
Finally, we believe that the distribution of solute atoms at and close to GBs which was obtained in the present work depending on the atomic radius of solute atoms might be qualitatively valid. Moreover, we are convinced that the present preliminary results might bring new qualitative insights on the connection between solute atom size and GB segregation. One promising option to improve our model is the introduction of more sophisticated LR potentials for iron, in order to reproduce accurately the elastic constants of this material.
As for the sentence “Actually, this issue is also a usual problem in ab-initio calculations, despite the accurate calculations the energy of system.”, it was indeed controversial, and it was removed from the manuscript.
[1] Lejček, P., Šandera, P., Horníková, J., Řehák, P., & Pokluda, J. (2017). Grain boundary segregation of elements of groups 14 and 15 and its consequences for intergranular cohesion of ferritic iron. Journal of Materials Science, 52(10), 5822-5834.
-Next, is is indeed so that the published range of Zener ratio is 1.5 < AZn < 2.7 ? Well, that is indeed a very wide range. By the way, why to call the Zener ratio A_Zn (when Zn is a chemical element!!)? It can be just A_Z without any possible confusion …
First, following the referee’s recommendation, the notation of Zener coefficient was changed to A_Z. Also, this range for Zener anisotropy coefficient was indeed reported in the literature. However, it was mentioned in the new version of the manuscript that the corresponding references based on both numerical and experimental values. In particular, A_Z=1,477 in [42] (numerical), A_Z=2,333 in [43] (numerical), and A_Z=2,698 in ref [A] below (experimental). For this reason, we believe that the presently obtained value for the parameter A_Z is satisfying, especially considering the simple interaction potential used in this work.
[A] Lide D R (ed) 2004Handbook of Chemistry and Physics85th edn (Boca Raton, FL: CRC Press)
-In general, the first letter of chemical elements, like iron or carbon, are not capitalized (not "Iron" and "Carbon") despite of the fact that their symbols are capitalized (i.e., Fe, C). Please correct throuhout the whole manuscript.
This mistake was corrected throughout the manuscript.
-Next, if the authors set the volume and other parameters of segregating elements “X1, X2 or X3” so that they are matching properties of specific elements such as tin, antimony or sulphur, I strongly suggest that this match to specific elements is mentioned in the Conclusions where the segregation behavior of these elements is discussed. What is this study good for when I only learn that "element X1" is diffusing faster than an "element X2"?
The conclusion was amended in this way. It is now mentioned in the conclusion that X_1, X_2 and X_3 solute atoms have a similar atomic radius to P Sb and Sn respectively.
-Lastly, as a minor detail, I suggest to call “the supercalculator CRIANN of Normandy” rather a “supercomputer” than a “supercalculator”.
This was also changed in the new version of the manuscript.
Reviewer 4 Report
The authors use a quasiparticle atomistic approach to study solute segregation at grain boundaries in alpha-Fe.
The research is timely and should be published in Materials subject to the following minor corrections:
(1) Line 5: "We this, we could ..." There is something missing here. Authors please amend.
(2) Line 27: "bore" should read "boron"
(3) Line 66: "describingf" should read "describing"
(4) Line 94: "qsystem" should read "system"
(5) Lines 142 and 143: k_B and T do not appear explicitly in Equation 1 (but rather in Equation 3). This should be amended.
(6) Line 252: "should were reckoned" There is something missing here.
(7) Line 293: "inucing" should probably read "inducing"
(8) Line 416: "significIntersetting" Please amend.
(9) Line 417: "ant segregation" Please amend.
(10) Line 454: "encured" Please amend.
(11) Line 487: "This consistent" should read "This is consistent"
Author Response
We would like to thank the reviewer for reviewing in details the present paper. Following the referee’s recommendation, a thorough proofread of the manuscript was performed. In particular, the 11 minor corrections required by the referee were done. More generally, typographical errors and mistakes in the English grammar and usage were systematically corrected. With this, we believe that the manuscript reads much better now.
Reviewer 5 Report
The authors proposed to use the QA to study the segregation of solute atoms at symmetric <100> tilt GBs in α-Fe. Three types of the solute atoms were considered and simulated carefully.
- The comparison between your simulation results and experimental results were not accomplished. Is there any laboratory findings by other researchers ?
- Fig.2 and L169-170 : You selected the step function shown in figure 2. Explain the reason why you chose this function as.
- Fig. 3 : Describe the error of θ in (a) and W in (b).
Author Response
We would like to sincerely thank the referee for reviewing the present manuscript. The following concerns raised by the reviewer were addressed:
- The comparison between your simulation results and experimental results were not accomplished. Is there any laboratory findings by other researchers ?
Unfortunately, only a very few number of results can be found on the atomic segregation pattern at GBs, especially when focusing on [100] tilt GB in alpha Iron. In this context, the vast majority of studies on the topic are numerical rather than experimental. As was mentioned in the paper, a good correspondence between the present simulations and previous DFT calculations from Lejvcek et al was observed. In addition, the formation of three dimensional capped trigonal prisms structure for Fe_9X clusters where X refers to phosphorous and boron solute atoms was observed could be simulated using molecular dynamics (see [48] in the article). This peculiar structures at GBs could also be reproduced by the present work.
From the experimental perspective, the segregation tendency of solute atoms at GBs is frequently addressed through the measurement of the enrichment factor of solute atoms at the GB using APT observations. However, this experimental technique cannot provide the exact position of atoms at GBs. Then, the experimental observation of the precise distribution of solute atoms in the vicinity of the GB remains particularly challenging.
[1] Lejček, P., Šandera, P., Horníková, J., Řehák, P., & Pokluda, J. (2017). Grain boundary segregation of elements of groups 14 and 15 and its consequences for intergranular cohesion of ferritic iron. Journal of Materials Science, 52(10), 5822-5834.
-Fig.2 and L169-170 : You selected the step function shown in figure 2. Explain the reason why you chose this function as.
Here, following the referee’s recommendation, the justification for the functional form of short range (SR) interactions was further justified in the manuscript. It was emphasized that this functional form for the SR contribution produces the spontaneous condensation of fratons into atomic spheres on the one hand, and prevents the overlap or atoms on the second hand. The introduction of SR interactions to the potential brings much more flexibility to the QA and allows to set different atomic radius for different atomic species.
-Fig. 3 : Describe the error of θ in (a) and W in (b).
On this point, we do not understand the recommendation of the referee. Indeed, no error can be defined in Fig 3, as this figure shows the potential used in this work. In details, Fig 3a gives the SR interaction potential in Fourier space, for the 4 different types of atoms considered in the present work: Fe, X_1 (same radius as P), X_2 (same radius as Sb) and X_3 (same radius as Sn). Then Fig3b) provides LR interactions in Fourier space.